# Learning Interaction-aware 3D Gaussian Splatting for One-shot Hand Avatars

**Xuan Huang**[1*], **Hanhui Li**[1*], **Wanquan Liu**[1], **Xiaodan Liang**[1],
**Yiqiang Yan**[2], **Yuhao Cheng**[2], **Chengqiang Gao**[1†]
[1]Shenzhen Campus of Sun Yat-Sen University
[2]Lenovo Research
huangx355@mail2.sysu.edu.cn lihh77@mail.sysu.edu.cn
liuwq63@mail.sysu.edu.cn xdliang328@gmail.com
yanyq@lenovo.com chengyh5@lenovo.com
gaochq6@mail.sysu.edu.cn

## Abstract

In this paper, we propose to create animatable avatars for interacting hands with 3D Gaussian Splatting (GS) and single-image inputs. Existing GS-based methods designed for single subjects often yield unsatisfactory results due to limited input views, various hand poses, and occlusions. To address these challenges, we introduce a novel two-stage interaction-aware GS framework that exploits cross-subject hand priors and refines 3D Gaussians in interacting areas. Particularly, to handle hand variations, we disentangle the 3D presentation of hands into optimization-based identity maps and learning-based latent geometric features and neural texture maps. Learning-based features are captured by trained networks to provide reliable priors for poses, shapes, and textures, while optimization-based identity maps enable efficient one-shot fitting of out-of-distribution hands. Furthermore, we devise an interaction-aware attention module and a self-adaptive Gaussian refinement module. These modules enhance image rendering quality in areas with intra- and inter-hand interactions, overcoming the limitations of existing GS-based methods. Our proposed method is validated via extensive experiments on the large-scale InterHand2.6M dataset, and it significantly improves the state-of-the-art performance in image quality. Project Page: https://github.com/XuanHuang0/GuassianHand.

## 1 Introduction

Recent advancements in 3D reconstruction and differential rendering techniques have significantly improved hand avatar creation and related applications. However, creating avatars for interacting hands from a single image remains challenging. The limited input view does not provide sufficient geometry and texture information for accurate reconstruction. Moreover, intra- and inter-hand interactions exacerbate information loss and introduce complex geometric deformations.

Extensive efforts have been made to tackle these issues, as shown in Figure 2: (a) Early approaches depend on explicit parametric meshes (e.g. MANO [1]) for geometry modeling, and utilize UV map [2, 3, 4, 5], vertex color [6, 7], or image space rendering [8] for appearance. Despite the efficiency in rendering, these methods fail to achieve realistic rendering results with the coarse mesh resolution and the simple combination of hand appearance and geometry. (b) More recently,

---

[*]Both authors contributed equally.
[†]Corresponding author.

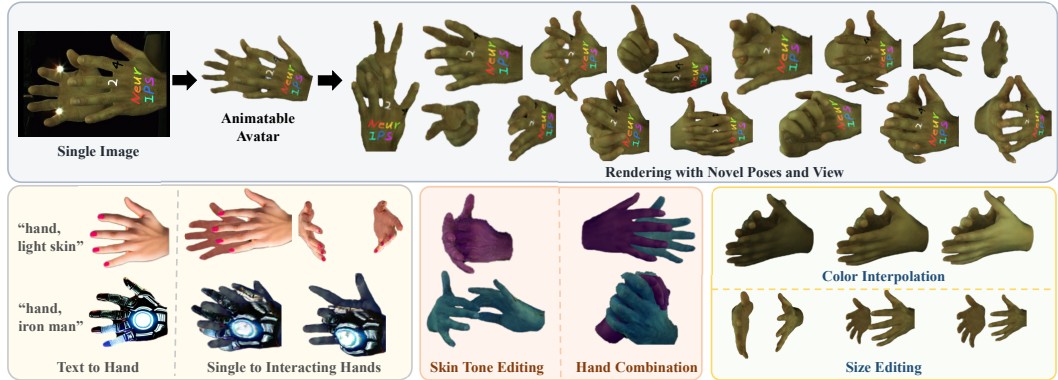

Figure 1: We present a novel interaction-aware Gaussian splatting framework that creates animatable interacting hand avatars from a single image. These high-fidelity avatars support various applications, such as editing, animation, combination, duplication, re-scaling, and text-to-avatar conversion.

with the significant success of neural radiance fields (NeRF), extensive studies [9, 10, 11, 12, 13] have employed NeRF-based models for implicit modeling. These methods [10, 11, 12] usually require per-scene optimization for each new identity using densely calibrated images, which results in expensive training costs. Generalizable NeRFs [14, 15, 13, 16] get rid of per-scene training by leveraging image-aligned features to enable reconstruction from a few or even a single view. Yet their dependence on image-aligned features also limits their performance under large view or pose variations. Besides, (c) One-shot NeRF-based methods [17, 18] propose to exploit data-driven priors with condition optimizations [17] and inversions [18]. Nevertheless, these methods are not suitable for our task, as they do not include any module to detect and handle interactions. Moreover, the inversion of identity vectors used in [18] omit the spatial image structure, which not only hinders its performance but also introduces extra time consumption for fine-tuning networks.

To tackle the above issues in existing methods, we aim to create animatable avatars for interacting hands with 3D Gaussian Splatting (GS) and single-image inputs. To this end, we introduce a novel two-stage interaction-aware GS framework as shown in Figure 2 (d). We disentangle the 3D presentation of hands into (i) features that can be effectively captured by training networks in the first stage of our framework (e.g., geometric features and latent neural texture maps), and (ii) identity maps that can be optimized efficiently in the per-subject one-shot fitting stage. In this way, our method not only enables leveraging cross-subject priors with learning-based features, but also well preserves per-subject characteristics via optimizing identity maps.

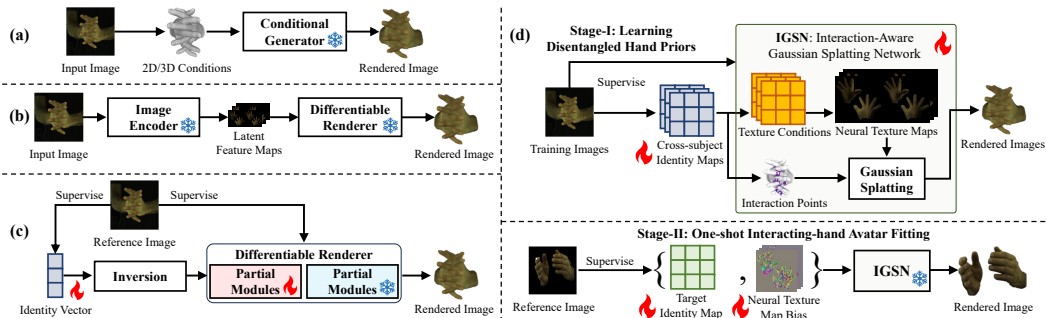

Figure 2: Paradigm comparison between existing one-shot hand avatar methods (a-c) and the proposed method (d). By decoupling the learning and fitting stages, our method leverages the advantages of learning-based methods (a, b) in modeling cross-subject hand priors, and the advantages of inversion-based methods (c) in one-shot fitting without the extra cost of network fine-tuning.

Additionally, to achieve robust reconstruction and enhance rendering quality, we devise an interaction-aware attention module and a self-adaptive Gaussian refinement module. The former module identifies Gaussian points with potential intra- and inter-hand interactions to enhance their features

with attention mechanisms. This enhancement allows our GS network to better model geometric deformations and fine-grained textures caused by interactions (e.g., wrinkles). The latter module is introduced to address the limitations of the coarse geometry of parametric hand meshes, which is achieved by learning to eliminate redundant Gaussians and assigning extra Gaussians in regions with complex textures and deformations. Consequently, our method can reconstruct realistic inter-hand avatars with great flexibility for animation and editing, as shown in Figure 1.

Overall, our contributions can be summarized as follows:

• We propose a novel two-stage interaction-aware GS framework to create animatable avatars for interacting hands from single-image inputs. Our method generates high-fidelity rendering results and supports various applications. Experimental results on the large-scale Interhand2.6M dataset [19] validate the superior performance of our method compared to previous methods.

• We disentangle the 3D presentation of hands into learning-based features that can be generalized well to different subjects and identity maps that are individually optimized for each subject. This disentanglement provides us with flexible and reliable priors for poses, shapes, and textures.

• We introduce an interaction-aware attention module, which identifies intra- and inter-hand interactions and further exploits interaction context to improve rendering quality.

• We devise a Gaussian refinement module that adaptively adjusts the number and positions of 3D Gaussian, which results in rendered images of higher quality under various hand poses and shapes.

## 2 Related Work

**One-shot Human Reconstruction**. One-shot reconstruction of 3D humans is a challenging and long-standing problem due to the limited information from a single input image. To alleviate this limitation, previous works leveraged parametric models [20, 1] as coarse geometry prior. Traditional methods [21, 22, 23, 24, 25] utilized UV maps for human appearance representation. To complete the unseen texture from the single image, some approaches [25, 24] inpainted missing textures via pre-trained diffusion models. Neural Radiance Fields (NeRF) [26] have also been explored for reconstruction from sparse views [27, 15, 14, 28] or a single view [29, 16, 13, 18]. KeypointNeRF [15] encoded spatial information using 3D skeleton keypoints. SHERF [16] created 3D human avatars from a single image with hierarchical features for informative encoding. VANeRF [13] leveraged visibility in both feature fusion and adversarial learning for single-view interacting-hand image synthesis. These methods greatly enhance the NeRF one-shot reconstruction performance. Nonetheless, generalizable NeRF [27, 30, 31, 14, 15] fails to achieve satisfactory results when the input image information is not sufficient for novel view prediction due to the under-utilization of hand priors. To overcome this issue, the pioneering research in [18] enabled one-shot single-hand avatar creation by learning data-driven hand priors which are further utilized with inversion and fitting. Compared with previous methods, our approach further disentangles the presentation of hand priors into latent geometric features, neural texture maps, and optimizable identity maps to enhance rendering quality and reduce the cost of one-shot fitting. Moreover, we introduce the interaction-aware module and self-adaptive refinement module, which helps to significantly improve the visual quality of synthesized images.

**Animatable Hand Avatar**. Conventional methods created hand avatars by incorporating UV textures with explicit parametric hand models. HTML [2] is the first parametric texture model of human hands which models hand appearance with several dimensions of variability. HARP [3] further introduced albedo and normal information into UV maps to represent hand appearance without any neural components. Handy [5] realistically captured high-frequency detailed texture using a GAN-based texture model. However, the rendering quality of these methods is constrained by the coarse geometry and sparsity of the MANO mesh. The recent advancements in the neural radiance field have resulted in the development of approaches that utilize implicit representation for hand reconstruction. LISA [9] is the first method that employs NeRF to learn the implicit shape and appearance of hands. HandAvatar [11] developed a high-resolution variant of MANO to fit personalized hand shapes and further disentangled the implicit representations for hands into geometry, albedo, and illumination. HandNeRF [10] designed a pose-driven deformation field with pose-disentangled NeRF to reconstruct single or interacting hands from multi-view images. LiveHand [12] proposed a low-resolution rendering of NeRF together with a super-resolution module to achieve real-time performance.

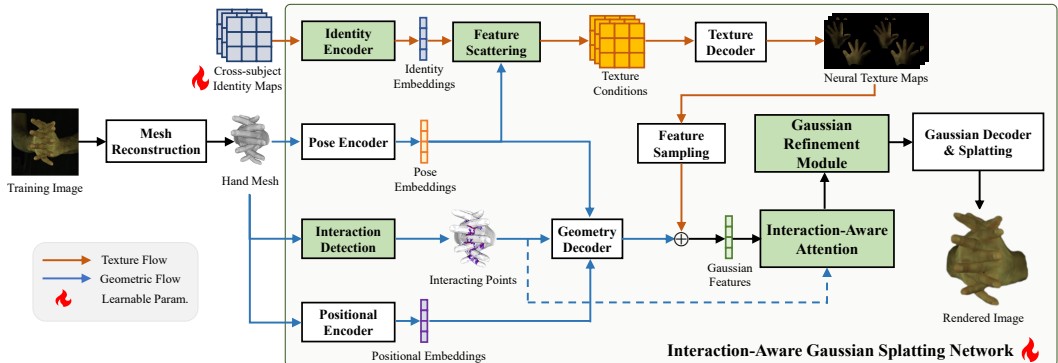

Figure 3: The architecture of the proposed interaction-aware Gaussian splatting network, of which the core components are the disentangled hand representation, the interaction detection module, the interaction-aware attention module, and the Gaussian refinement module labeled in green.

**Point-based 3D Representation and Rendering**. There has been a growing interest in point-based neural rendering due to recent advancements in 3D Gaussian splatting (3DGS) [32], which has led to high-quality and real-time rendering speed. Since 3DGS is primarily designed for static scenes, many efforts [33, 34, 35, 36] are dedicated to expanding its applicability to free pose animation and rendering. 3D-PSHR [34] achieved real-time and photo-realistic hand reconstruction from large-scale multi-view videos based on 3D points splatting. Our method differs from 3D-PSHR as it allows instant single-view one-shot hand avatar reconstruction and saves the expensive computational cost of per-scene optimization.

## 3 Methodology

In this section, we introduce the details of the proposed two-stage framework for creating interacting hand avatars from a single image. The key ideas of our framework contain three aspects: (i) We address the lack of information caused by limited inputs by learning disentangled priors for hand poses, shapes, and textures (Sec. 3.1). (ii) We construct an interaction-aware Gaussian splatting network to handle both intra- and inter-hand interactions (Sec. 3.2). (iii) Leveraging invertible identity and neural texture maps, we reduce the time consumption of one-shot avatar reconstruction while simultaneously improving the quality of synthesized images (Sec. 3.3).

### 3.1 Disentangled 3D Hand Representation

To demonstrate the motivations of the proposed disentangled 3D hand representation, we first provide the formulation of our task.

**Task formulation**. Given a reference image of interacting hands $I_r$, our task is to reconstruct an animatable two-hand avatar that can generate images of the hands with novel poses and from novel views. To achieve this, we propose to construct a differentiable renderer $\mathcal{R} : \mathbb{R}^{|\theta| \times |c|} \to \mathbb{R}^{H \times W \times 3}$, where $\theta$ and $c$ denote the hand pose parameters and camera parameters, respectively. $H$ and $W$ denote the height and width of rendered three-channel RGB images.

More specifically, we propose to implement $\mathcal{R}$ via Gaussian splatting (GS) [32] for its advantages in explicit modeling and computational efficiency. Essentially, GS is a point-based rasterization technique that leverages a set of 3D points (Gaussians) with attributes like colors, opacity, and spherical harmonic coefficients to represent reference images. Let $\mathcal{P} \in \mathbb{R}^{N \times 3}$ denote the set of $N$ Gaussians and $\mathbf{a} \in \mathbb{R}^{N \times D}$ denote their D-dimensional attributes. To preserve the characteristic of the hands in $I_r$, we need to optimize $\mathcal{R}$ via minimizing the following $l_1$ loss,

$$\underset{\mathcal{P}, \mathbf{a}, \theta, c}{\arg\min}(\|\mathcal{R}(\theta, c \,|\, \mathcal{P}, \mathbf{a}) - I_r\|_1). \tag{1}$$

Due to the high-dimensional attribute space and inter-attribute interference, direct optimization for Eq. (1) is intractable. Although learning-based methods can alleviate this issue to a certain extent, their

results may fall short if $I_r$ lies outside of the distribution (OOD) of their training data. Therefore, a disentangled representation that integrates optimization-based and learning-based Gaussian attributes is essential for addressing our task effectively. Furthermore, a reliable representation should capture both the geometric and textural properties of hands. Given the availability of extensive hand mesh reconstruction methods for geometric information, we propose leveraging learning-based geometric properties alongside optimizable textures in our disentangled representation. This approach allows us to handle the diversity and potential OOD issues of hand textures.

To this end, we devise the disentangled representation shown in Figure 3 to combine explicit geometric embeddings from hand meshes with neural texture maps encoding implicit latent fields. The encoders for this representation are learned on a training dataset consisting of images of $S$ subjects. Specifically, we first construct the optimizable cross-subject identity maps $\mathbf{m} \in \mathbb{R}^{S \times 2C \times H \times W}$, where $C$ denotes the number of feature channels of one hand. For each training image $I_t$, we reconstruct a parameterized hand mesh $\mathcal{M}$ from it and use the vertices of $\mathcal{M}$ to initialize $\mathcal{P}$. Let $s \in [1, .., S]$ denote the subject ID of $I_t$, we retrieve its corresponding identity map $\mathbf{m}_s$ and combine it with the pose embedding from $\mathcal{M}$ to infer its neural texture map $\mathbf{t}_s \in \mathbb{R}^{2C \times H \times W}$. Consequently, $\forall p \in \mathcal{P}$, we can query their feature vectors from $\mathbf{t}_s$ based on their texture coordinates and predict their Gaussian attributes. In the one-shot fitting stage, to obtain the neural texture map of $I_r$, we only need to optimize a new identity map initialized with zeros. This is applicable, as our identity maps and natural texture maps share an important advantage: they both preserve the spatial structure of textures, which overcomes the burdens of previous vector-based inversion methods [18, 37].

Below we introduce the core components for implementing our disentangled representation. These components will be integrated into the GS network in the next section for end-to-end learning.

**Parameterized Hand Mesh**. We employ MANO [1] to reconstruct hand meshes from images for its convenience in animation. MANO is a parametric model that represents a hand mesh by pose parameters $\theta \in \mathbb{R}^{48}$ and shape parameters $\beta \in \mathbb{R}^{10}$. To further improve the mesh quality, we use a high-resolution version of MANO [11]. We follow [12] to obtain normalized UV coordinates $(u, v) \in [0, 1] \times [0, 1]$ of mesh vertices and project them onto the neural texture plane.

**Geometric Encoding**. To exploit explicit geometric features from hand meshes, we utilize a pose encoder and a positional encoder. The pose encoder is an MLP taking the pose parameters $\theta$ and the camera parameters $c \in \mathbb{R}^{25}$ as inputs (which is the flattened concatenation of an extrinsic matrix $\in \mathbb{R}^{4 \times 4}$ and an intrinsic matrix $\in \mathbb{R}^{3 \times 3}$). Our pose embeddings do not involve $\beta$ and hence they are independent of identities. The positional encoder is a shallow PointNet [38] with local pooling [39]. We further employ a transformer-based decoder to merge the outputs of the pose encoder and the positional encoder, similar to [40].

**Texture Encoding**. Given the UV coordinates of a vertex, we retrieve its feature vector on the optimizable identity map, along with the $\gamma$ positional encoding [26] of its coordinates to generate its identity embedding. The identity embeddings of all vertices are concatenated with the pose embedding and projected (scattered) back to the UV plane to form a texture condition map. We again adopt a transformer-based decoder to process the condition map and yield the neural texture map.

Finally, we combine the geometric feature vectors of all vertices with their texture feature vectors using element-wise addition. This results in a unified latent representation $\mathbf{f} \in \mathbb{R}^{|\mathcal{P}| \times C}$, which we use for predicting Gaussian attributes.

## 3.2 Interaction-Aware Gaussian Splatting Network

To better reconstruct interacting hand avatars with various poses, we propose to enhance the Gaussian features $\mathbf{f}$ via an interaction-aware attention (IAttn) module and a Gaussian point refinement module (GRM). IAttn identifies potential points with intra- or inter-hand interaction. By exploring the context around interaction points, IAttn improves the reconstruction quality of geometric deformations and texture details resulting from interactions (such as shading, wrinkles, and veins). Furthermore, GRM not only eliminates redundant Gaussians but also generates additional Gaussians near regions with complex textures. With these two modules, our network can render high-quality hand images with rich details.

### 3.2.1 Interaction-aware Attention

To detect interacting points in $\mathcal{P}$, we propose a straightforward yet effective strategy that calculates the difference between the neighboring point sets of posed hand meshes and a canonical mesh. This strategy is practical as we can define an interaction-free mesh as the canonical one. For an arbitrary query point $q \in \mathcal{P}$, if its top-$N_c$ nearest neighboring points on the canonical mesh $\Omega_c(q)$ are significantly overlapped with its top-$N_p$ nearest neighbors on the posed mesh (denoted as $\Omega_p$), the chance that $q$ is an interacting point is low. This strategy can be formulated as follows,

$$d(q) = \begin{cases} 1, & \text{if } |\Omega_c(q) \cup \Omega_p(q) - \Omega_c(q) \cap \Omega_p(q)| > T, \\ 0, & \text{otherwise,} \end{cases} \tag{2}$$

where $T$ is a user-defined threshold. Additionally, we append the interacting label $d(q)$ to the pose embedding introduced in the last section. Note that our proposed strategy can detect both self-interacting and cross-interacting points. To maintain the efficiency of our method, we only conduct self-attention on detected interacting points.

### 3.2.2 Self-adaptive Refinement for 3D Gaussians

Gaussians initialized from MANO can only provide coarse hand geometry with restricted deformations. To better model the geometry of hands with various poses and shapes, we devise the self-adaptive GRM to control the density of Gaussians and refine their locations. Given a Gaussian point $p$ and its corresponding feature vector $\mathbf{f}_p$, we utilize an MLP $\phi(\mathbf{f}_p)$ with the sigmoid activation to predict the validity of $p$, i.e., $\phi : \mathbb{R}^C \to [0, 1]$. We remove $p$ from $\mathcal{P}$ if $\phi(\mathbf{f}_p)$ is below a pre-defined threshold $T_d$ while splitting $p$ if $\phi(\mathbf{f}_p)$ is larger than another threshold $T_s$. We also exploit $\mathbf{f}$ to predict the offsets of Gaussians to adjust their positions.

### 3.2.3 Network Optimization

With the search space of Gaussian attributes reduced significantly by the proposed disentangled representation, we are now ready to optimize our GS network and learn cross-subject hand priors. We train the GS network along with the optimizable cross-subject identity maps $\mathbf{m}$ by minimizing the following loss:

$$I = \mathcal{R}(\theta, c, \mathbf{m}, \mathcal{P}|\Theta), \quad \mathcal{L}_{rec} = \lambda_{rgb} \|I - I_t\|_1 + \lambda_{VGG} \mathcal{L}_{VGG}(I, I_t), \tag{3}$$

where $\Theta$ are the learnable parameters in our GS network. $\mathcal{L}_{VGG}$ denote the perceptual loss [41]. $\lambda_{rgb}$ and $\lambda_{VGG}$ are user-defined weights.

## 3.3 One-shot Hand Avatar Reconstruction

In the stage of one-shot hand avatar reconstruction, the parameters of IGSN are fixed and we fine-tune the identity map of the new subject $\mathbf{m}^* \in \mathbb{R}^{2C \times H \times W}$. This can be formulated as,

$$\underset{M, \mathbf{m}^*}{\arg\min} \mathcal{L}_{inv.} = \mathcal{L}_{rec}(\mathcal{R}(\theta_r, c_r, \mathbf{m}^*, \mathcal{P}_r|\Theta), I_r) + \lambda_{mask} \|M - M_r\|_2^2, \tag{4}$$

where the geometric parameters $\theta_r$, $c_r$, and $\mathcal{P}_r$ can be obtained via off-the-shelf MANO regressors [42, 43]. $M$ and $M_r$ denote the hand mask of $I$ and $I_r$, respectively. $\lambda_{mask}$ is the empirical weight of the loss term on masks.

[18] suggests that optimization tricks like color calibration and view regularization can further improve the synthesized images. Inspired by this, we also introduce a texture map bias $\Delta \mathbf{t} \in \mathbb{R}^{C \times H \times W}$ to modulate the latent neural feature maps $\mathbf{t}$ of two hands ($\mathbf{t} = \{\mathbf{t}_l, \mathbf{t}_r\}$). That is, $\mathbf{t}_l := \mathbf{t}_l + \Delta \mathbf{t}$ and $\mathbf{t}_r := \mathbf{t}_r + \Delta \mathbf{t}$. We assume that $\Delta \mathbf{t}$ can be shared by $\mathbf{t}_l$ and $\mathbf{t}_r$ due to the symmetry of the left and right hands. We include a regularization term on $\Delta \mathbf{t}$ into Eq. (4) to prevent drastic shifts of $\mathbf{t}$ as follows:

$$\underset{M, \mathbf{m}^*, \Delta \mathbf{t}}{\arg\min} (\mathcal{L}_{inv.} + \lambda_{reg} \|\Delta \mathbf{t}\|_2^2), \tag{5}$$

where the regularization weight $\lambda_{reg}$ is user-defined. In our experiments, we find that adding $\Delta \mathbf{t}$ accelerates the fitting process and helps to prevent undue changes.

Table 1: One shot synthesis comparison with state-of-the-art methods on Interhand2.6M.

| Method | Novel View Synthesis | | | Novel Pose Synthesis | | |
|---|---|---|---|---|---|---|
| | PSNR↑ | SSIM↑ | LPIPS↓ | PSNR↑ | SSIM↑ | LPIPS↓ |
| KeypointNeRF | 23.55 | 0.804 | 0.326 | - | - | - |
| SMPLpix | 24.50 | 0.868 | 0.170 | 24.26 | 0.854 | 0.173 |
| VANeRF | 25.38 | 0.848 | 0.226 | 24.42 | 0.822 | 0.250 |
| OHTA* | 25.31 | 0.851 | 0.184 | 25.93 | 0.880 | 0.156 |
| Ours | **26.14** | **0.869** | **0.161** | **26.56** | **0.890** | **0.133** |

## 4 Experiments

### 4.1 Setup

**Learning IGSN.** Our experiments are conducted on the publicly available Interhand2.6M dataset [19] (CC-BY-NC 4.0 licensed) that consists of large-scale multi-view sequences of different subjects performing various hand poses. Following [18], we adopt interacting-hands pose sequences of 21 subjects from InterHand2.6M training set for pre-training. For each subject, an unseen pose sequence is used for evaluation. Our network is trained on three A6000 GPUs using the Adam optimizer [44] with the learning rates of $1 \times 10^{-4}$ for eight epochs. Loss weights in Eq. (3) are set as $\lambda_{rgb} = 10.0, \lambda_{VGG} = 0.1$. For interaction detection, we set $N_c = 100$ and $T = 90$. For self-adaptive GRM, we set $T_d = 0.1$ and $T_s = 0.9$. We adopt a coarse-to-fine mesh refinement strategy during training: For the first 5 epochs, we upsample hand meshes to 12,337 points per hand while for the last three epochs, we further upsample hand meshes to 49,281 points per hand.

**One-shot Reconstruction.** We conduct one-shot reconstruction evaluations on the testing set of InterHand2.6M as in [18, 11]. To evaluate the novel pose rendering quality, We evenly sample 349 frames from four pose sequences including four common views in the "test/capture0" subset as the test set. To assess the quality of novel view synthesis, we have selected the initial 50 views from the "test/capture0" subset. The one-shot fitting takes 50 optimization steps with the learning rate of $1 \times 10^{-2}$. The whole process takes 2.5 minutes with an A6000 GPU. Loss weights in Eq. (4,5) are set as $\lambda_{mask} = 1.0, \lambda_{reg} = 0.01$.

**Baselines and Metrics.** We select four state-of-the-art methods for comparison. We adopt two generalizable NeRFs including KeypointNeRF [15] designed for human novel view synthesis and VANeRF [13] designed for single-view interacting-hand image novel view synthesis. We further adapt VANeRF to the one-shot animatable interacting hands reconstruction. Moreover, We include SMPLpix [8] as an image-space baseline. Besides, although OHTA [18] is not designed for interacting hands reconstruction, we still implement its one-shot strategy with our pre-trained model (denoted as OHTA*) for one-shot performance comparison. Following previous works [13, 8, 15, 18], we report LPIPS[45], PSNR[46], and SSIM[47] as the metrics of rendering quality.

### 4.2 Comparison with State-of-the-art Methods

**Quantitative Comparison.** Table 1 reports the quantitative results of our method against the baselines in the one-shot reconstruction scenario, including novel view synthesis and novel pose synthesis. We can see that our method significantly outperforms all methods on all metrics in both tasks. NeRF-based methods, KeypointNeRF and VANeRF fail to have good performance facing large view or pose variations due to the under-utilization of the hand priors. SMPLpix as an image-space method lacks generalization and 3D understanding when coping with single-view reconstruction. Compared with OHTA, our method captures more accurate characteristics of the target identity using the identity map and neural map bias.

**Qualitative Comparison.** Figure 4 demonstrates the visual comparison between our approach and the baselines. SMPLpix fails to produce a reasonable hand appearance with the limited information from a single image. VANeRF predicts the basic hand geometry while leaving high-frequency details like wrinkles and veins. OHTA recovers a reasonable hand geometry with most of the appearance close to the target identity while failing to capture fine-grained identity features. Compared with baselines, our method successfully recovers hand details (e.g. nails, wrinkles, and veins) of the

| GT | SMPLpix | VANeRF | OHTA* | Ours | GT | SMPLpix | VANeRF | OHTA* | Ours |
|----|---------|--------|-------|------|----|---------|--------|-------|------|

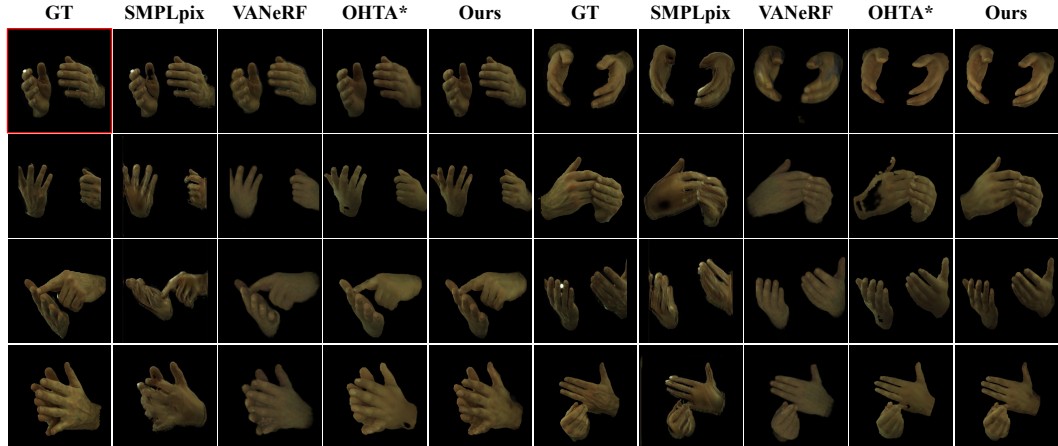

Figure 4: Qualitative comparisons with state-of-the-art methods. The input image is shown in the top-left grid labeled in red. The first row presents results without changing the pose from the input view (left) and an alternative view (right), while results in the remaining rows are with novel poses.

target identity for various views and poses. The qualitative comparisons demonstrate our robust performance for one-shot animatable hand avatars creation.

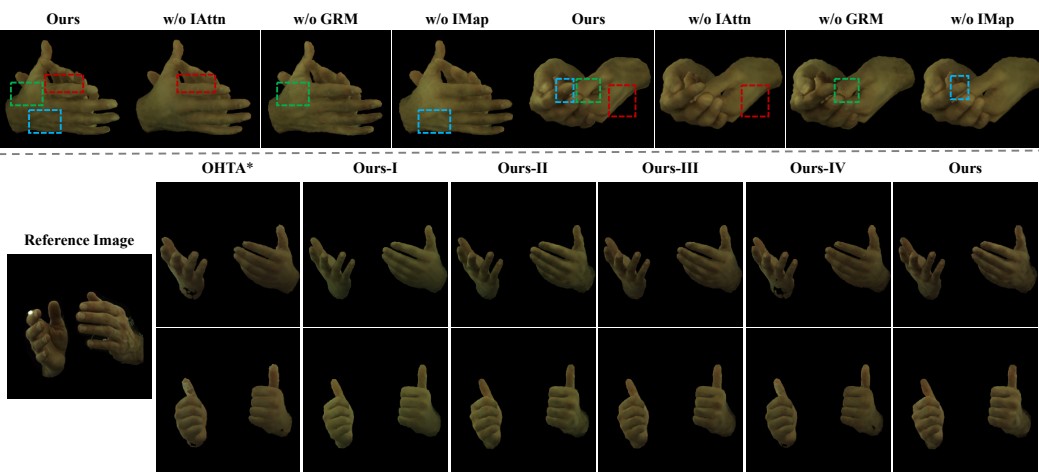

Figure 5: Visual examples of the ablation study on the proposed components in the hand-prior learning stage (top) and the one-shot fitting stage (bottom).

## 4.3 Ablation Study

We conduct ablations studies in each of the two stages of our framework. Particularly, we focus on the components of IGSN in the first stage while the loss terms for one-shot fitting in the second stage.

### 4.3.1 Interaction-aware Gaussian Splatting

The evaluation is conducted in the "train/capture0" subset with 23 pose sequences for training and 1 for testing. The quantitative results are reported in Table 2 Stage-One. The results reflect that each design does bring performance gains and the best performance is obtained by the full model.

**Effectiveness of GRM.** To validate the effectiveness of our Gaussian points refinement module, we implement a variant by substituting refined Gaussian points with upsampled hand mesh points (denoted as w/o GRM. in Table 2). To further verify the effectiveness of our points modification based on the points validation prediction, we implement a variant by replacing the Gaussian points

Table 2: Ablation study on the components in our two-stage framework.

| Stage-One | | | | Stage-Two | | | | | | | |
|---|---|---|---|---|---|---|---|---|---|---|---|
| Method | PSNR↑ | SSIM↑ | LPIPS↓ | Method | IMap | Inv | Calib | MBias | PSNR↑ | SSIM↑ | LPIPS↓ |
| | | | | OHTA* | | √ | √ | | 25.93 | 0.880 | 0.156 |
| w/o IMap. | 27.76 | 0.900 | 0.139 | Ours-I | √ | | | | 26.32 | 0.889 | 0.150 |
| w/o IAttn. | 27.52 | 0.898 | 0.143 | Ours-II | √ | √ | | | 26.50 | 0.888 | 0.141 |
| w/o GRM. | 27.24 | 0.893 | 0.177 | Ours-III | √ | √ | √ | | 26.50 | 0.887 | 0.139 |
| Ours | **28.11** | **0.902** | **0.130** | Ours-IV | | √ | √ | √ | 26.14 | 0.883 | 0.145 |
| | | | | Ours | √ | √ | √ | √ | **26.56** | **0.890** | **0.133** |

refinement module with simply predicting offset for each point (denoted as w/o GPM. in Figure 5). The results indicate that the points modification eliminates the redundant points and densifies the detailed area based on point features, producing better rendering results.

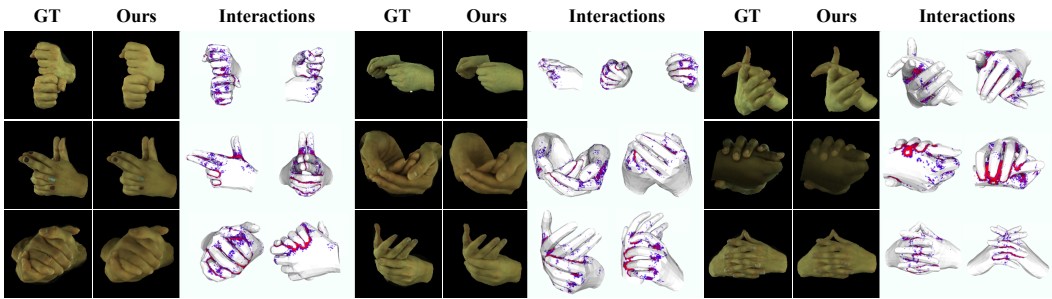

Figure 6: Visualization of intra- and inter-hand interactions detected by the proposed method. Areas with interactions from sparse to dense are labeled in colors from blue to red, respectively.

**Effectiveness of IAttn.** Figure 6 demonstrates the predictions of the proposed interaction detection in IAttn. We can observe that our designed interaction detection effectively recognizes the interaction areas, including one-hand self-interaction and the interaction between two hands.

To validate the effectiveness of the interaction-aware attention module, we remove it from our full model (denoted as w/o IAttn. in Table 2 and Figure 5). The results show that the proposed module improves the reconstruction quality of geometry deformation and texture details caused by interaction based on effective interaction detection.

**Effectiveness of Identity Map.** We also evaluate the effectiveness of the identity map by substituting it with the identity code (vector) as in [18] (denoted as w/o IMap. in Table 2 and Figure 5). The results demonstrate that the Identity Map contributes to high-fidelity hand reconstruction by capturing the fine-grained texture of the target identity.

### 4.3.2 One-shot Fitting

We show the effectiveness of our strategies for one-shot reconstruction in Table 2 Stage-Two and Figure 5. When the identity map is replaced with identity code (IV), the performance drops indicating that our identity map can capture more accurate and fine-grained characteristics of the target hand. When there is no neural texture map bias (III), the details of the target hands are missing. Without color calibration (II), the results become worse due to the color bias. When omitting the identity inversion (I), a significant drop in LPIPS is observed as the target identity information is not captured. Overall, we validate the effectiveness of our design for one-shot reconstruction.

## 5 Conclusion

We tackle the challenging single-image interacting hand avatar reconstruction task via an interaction-aware Gaussian splatting framework in this paper. Our framework disentangles 3D hand representations into learning-based features that can be extracted by the trained network and identity maps

that are one-shot optimized on the hands of new subjects. Additionally, our framework employs an interaction-aware attention module and a self-adaptive refinement module to detect and handle regions with intra- and inter-hand interactions. The proposed method outperforms cutting-edge methods on the Interhand2.6M dataset and creates high-quality avatars for various tasks successfully.

## 6    Acknowledgments

This work was supported in part by National Key R&D Program of China (2022YFA1004100), National Science Foundation of China Grant No. 62176035, No. 62372482, No. 62476293, and No. 61936002, National Science and Technology Major Project (2020AAA0109704), Guangdong Outstanding Youth Fund (Grant No. 2021B1515020061), Shenzhen Science and Technology Program (Grant No. GJHZ20220913142600001), Nansha Key RD Program under Grant No.2022ZD014.

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

# A  Appendix

## A.1  Application.

As shown in Figure 7, We demonstrate more visual results of various applications, including in-the-wild results from real-captured images, text-to-avatar, and texture editing. For text-to-avatar, we utilize ControlNet [48] with depth maps as condition information for hand image generation. For in-the-wild results, we use ACR [43] to estimate hand pose and camera parameters from real-captured images. These results clearly show that our method can be applied to in-the-wild images and obtain considerable results.

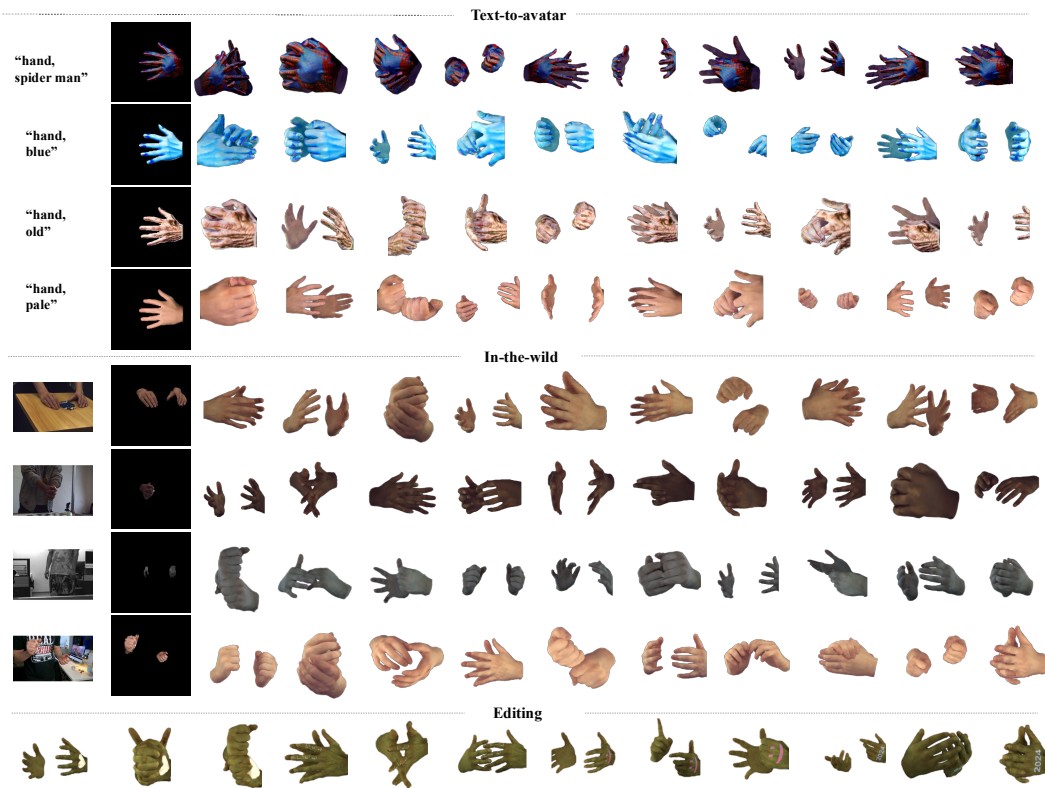

Figure 7: Visual examples of the proposed method in various scenarios, including text-to-avatar, in-the-wild reconstruction from real images, and texture editing.

## A.2  Statistical Significance

To evaluate the statistical significance, we run the proposed method five times with different random initiation of identity map in the second stage. Table 3 presents the mean and standard deviation of the results.

Table 3: Mean and standard deviation of the performance of the proposed method.

| Method | PSNR↑ | SSIM↑ | LPIPS↓ |
|--------|-------|-------|--------|
| Ours | 26.6027±0.0237 | 0.8879±0.0009 | 0.1342±0.0008 |

## A.3  Visualization of GRM

Figure 8 presents the qualitative comparisons of hand geometry and rendering results between our Gaussian refinement modules and vanilla GS with mesh upsampling. We can see that, the hand points initialized from hand parameters can only provide approximate hand geometry while refined Gaussian points excel in producing more detailed texture and realistic hand geometry with the help of the Gaussian points Refinement module.

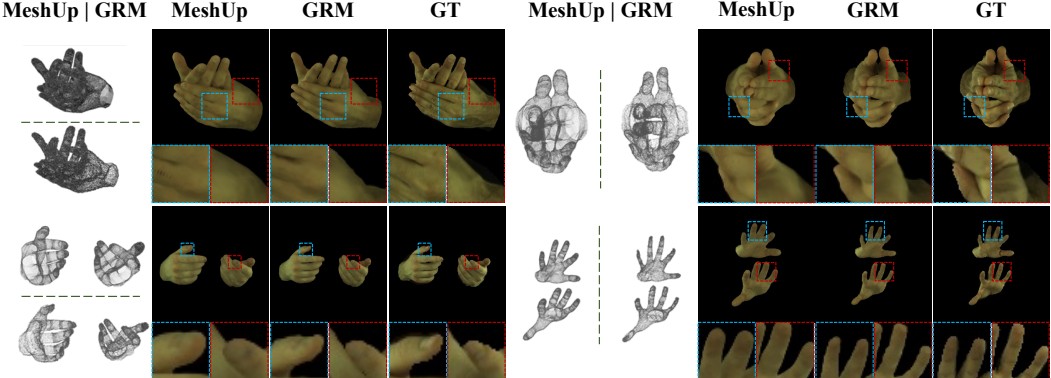

Figure 8: Qualitative comparisons between Gaussians by mesh upsampling (denoted as MeshUp) and the proposed GRM.

## A.4 Data Selection and Preprocessing

We use the same captures for training with OHTA [18] in stage one and save the pose sequence '0275_left-babybird' for testing. For the evaluation of stage two, we conduct experiments on 'test/Capture0' including four pose sequences 'ROM01_No_Interaction_2_Hand', 'ROM01_No_Interaction_2_Hand', 'ROM09_Interaction_Fingers_Touching', and 'ROM09_Interaction_Fingers_Touching_2'.

For image preprocessing, we crop out the hand region with the bounding boxes and resize the cropped area to $256 \times 256$ consistent with previous methods [13, 18]. We adopt SAM [49] to produce hand masks for better segmentation compared to MANO-rendered masks.

## A.5 More Ablations

Figure 9 demonstrates visual examples of shadow disentanglement (top) and qualitative results of four ablation studies (bottom). The quantitative results of the four ablation studies are provided in Table 4.

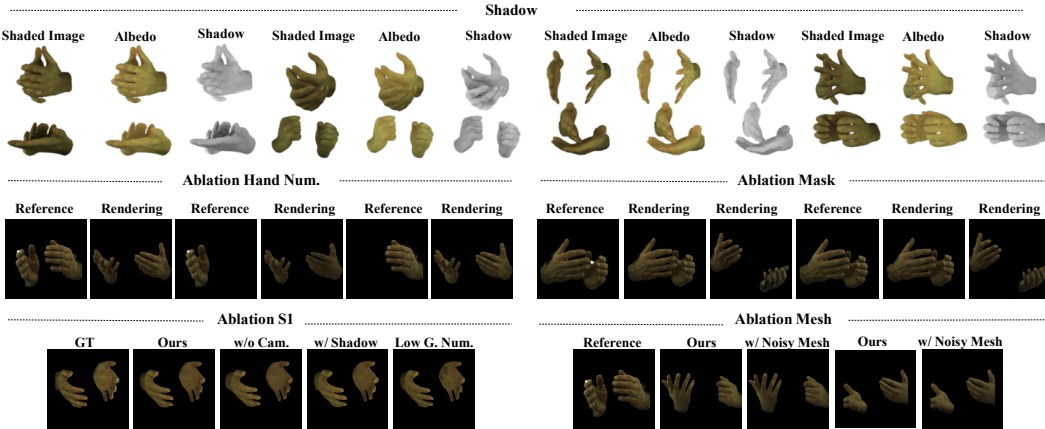

Figure 9: Visual examples of shadow disentanglement (top) and four ablation studies (bottom).

**Shadow Disentanglement (denoted as Shadow).** Figure 9 (top) demonstrates visual examples of shadow disentanglement including a shaded image, an albedo image, and a shadow image for each pair of hands. This figure clearly shows the shadow (dark) areas, which suggests that our method can disentangle albedo and shadow.

**Ablation study on single-hand images (denoted as Ablation Hand Num.).** The advantage of using images of both hands is that two-hand images can provide more complementary information compared with single-hand images, which helps improve the model's performance. We experiment to compare the performance between single-hand images and two-hand images by masking one of the hands from the reference image. The results show that our method achieves better performance compared with the single-hand baseline, which is reasonable as two-hand images contain more complementary information.

**Ablation study on segmentation method (denoted as Ablation Mask).** To validate the impact of segmentation masks, we consider masks predicted by SAM and the ground-truth meshes. We observe that SAM masks enhance the performance of our model, since they are better aligned with hands and prevent background clutters.

**Ablation S1.** Ablation S1 shows the visual results of three different ablation settings: including (i) the use of camera parameters, which is a variant of our method without the camera parameters (w/o Cam.) and suffers from obvious performance drop; (ii) adding shadow coefficient, which incorporates shadow coefficient prediction into the proposed method (w/ Shadow); (iii) Low Gaussian Points, which reduces the number of Gaussian points to 24k to show that coarsen hand boundaries are caused by fewer numbers of Gaussian points.

**Ablation study on mesh estimation method (denoted as Ablation Noise).** We use hand parameters estimated by ACR [43] to analyze how the accuracy of mesh reconstruction impacts the final results. The performance of our method with the predicted parameters is still satisfying, which suggests our method is robust to noise hand meshes to a certain extent.

Table 4: The quantitative results of four ablation studies.

| Ablation Hand Num. | | | | Ablation Noise | | | | |
|---|---|---|---|---|---|---|---|---|
| Method | PSNR↑ | SSIM↑ | LPIPS↓ | Method | Mesh | PSNR↑ | SSIM↑ | LPIPS↓ |
| Ours | **26.14** | **0.869** | **0.161** | Ours | GT | **26.14** | **0.869** | **0.161** |
| Ours-Right | 24.81 | 0.852 | 0.178 | Ours | ACR | 25.83 | 0.864 | 0.167 |
| Ours-Left | 25.58 | 0.858 | 0.172 | | | | | |
| w/o Cam. in Ablaiton S1 | | | | Ablation Mask | | | | |
| Method | PSNR↑ | SSIM↑ | LPIPS↓ | Method | Mask | PSNR↑ | SSIM↑ | LPIPS↓ |
| Ours | **28.11** | **0.902** | **0.130** | Ours | SAM | **26.52** | **0.888** | **0.135** |
| w/o Cam. | 25.91 | 0.862 | 0.198 | Ours | Mesh | 25.99 | 0.877 | 0.142 |

## A.6 Implementation of OHTA*

Since OHTA [18] as a single-hand reconstruction method cannot be directly applied to our scenario, we instead use our pre-trained model and only compare the design of the fine-tuning stage. We implement OHTA* with the texture inversion stage and texture fitting stage as described in its paper. In the texture inversion, we keep the network weights frozen and optimize the identity code along with per-channel color calibration coefficients to produce a similar appearance to the target identity of the input image. In the texture fitting, we fine-tune the texture feature MLP for texture feature extraction and constrain the texture-fitting results of some reference views to be close to the rendering results before texture-fitting.

## A.7 Limitations and Society Impact

Although we have greatly shortened the fine-tuning process, the per-identity training still limits the application compared to single forward inference. Moreover, separate optimization for each identity hinders the model from integrating similar identity information optimized in the fine-tuning stage.

When facing serious errors in pose estimation, our method may fail to model the proper appearance through the wrong alignment caused by misleading hand parameters as shown in Figure 10. Severe estimation errors inevitably cause degraded performance, which is also mentioned in the paper of OHTA [18].

Our method has a positive impact on society as it can facilitate sign language production by creating high-fidelity, animatable hand avatars with interaction. There are no negative societal impacts concerning our work.

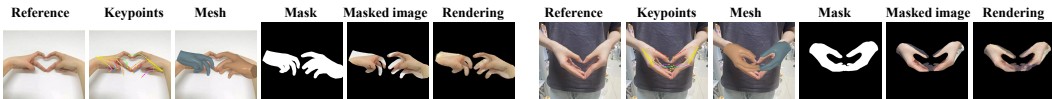

Figure 10: Visual examples of failure cases.

