# OpenReview forum: "Learning Interaction-aware 3D Gaussian Splatting for One-shot Hand Avatars"
_NeurIPS.cc/2024/Conference — NeurIPS 2024 poster_

### Official Review · Reviewer_mjNr · 2024-07-09

**Soundness:** 3
**Presentation:** 4
**Contribution:** 3
**Rating:** 6
**Confidence:** 3

**Summary:**

This paper proposes a method to create a 3D Gaussian Splatting (GS)-based avatar for interacting hands from single-image inputs. Its main contribution is to propose a two-stage GS framework (1) to leverage cross-identity priors via learning-based features and also to (2) well preserve per-identity information via optimized identity maps. Additionally, it proposes an interaction-aware attention module and a Gaussian refinement module to further exploit the hand interaction context for more realistic rendering. In the experiments, the proposed methods outperforms the existing methods based on generalizable and one-shot NeRFs.

**Strengths:**

**(1) Good presentation.** Overall, the paper is well-organized and technical explanations are sound. The figures are also clearly presented.

**(2) Sufficient technical novelty.** The disentanglement of hand representation into learning-based features and optimization-based identity maps sounds reasonable. Also, to the best of my knowledge, this is one of the first works on generating avatars for interacting hands from single image inputs.

**(3) Good experimental validation.** The paper reports sufficient ablation study results and performs adequate comparisons with the existing baselines.

**Weaknesses:**

**(1) Effects of off-the-shelf MANO regressor and mask detector.** The proposed avatar reconstruction method (Sec. 3.3) uses the outputs of off-the-shelf MANO regressor and segmentation model. It would be informative to include the discussions on how the quality of these outputs might affect the final rendering quality.

**(2) Non-smooth hand boundaries in rendering results.** In Fig. 4, I’ve noticed that the boundaries of hands rendered using the proposed method are less smooth compared to the existing methods. Does it indicate that the proposed GS-based method (implicitly) learns less smooth hand geometries than the other NeRF or mesh-based methods? More discussions regarding this would be informative.

**Questions:**

(1) In lines 94-95, the paper claims that the rendering quality of the existing MANO-based methods “is constrained by the coarse geometry and sparsity of the MANO mesh”. However, to my knowledge, HARP [3] and Handy [5] use higher resolution-version of MANO meshes to already address this issue. What would be the advantage of using GS compared to using such higher-resolution hand meshes?

(2) In lines 118-119, the paper states that the proposed method "reduce[s] the time consumption of one-shot avatar reconstruction". Have you tried comparing the reconstruction time of the proposed method against OHTA [18]?

**Limitations:**

The authors discussed the limitations in Sec. A.4 in Appendix.

---

> ### Author Rebuttal · Authors · 2024-08-06
>
> We sincerely thank the reviewer’s confirmation of sufficient technical novelty. To address the concerns of the reviewer, we conduct several ablation studies and provide extra visual examples in the RF. Our responses are listed as follows:
>
> **W1 Effects of off-the-shelf MANO regressor and mask detector**: Thank you for your advice. We add experiments accordingly as follows:
>
> (i) We experiment to evaluate the effects of the off-the-shelf MANO regressor (ACR). The qualitative results are shown in RF-Fig. 2 Ablaiton Mesh and the quantitative results are listed as follows:
>
> | Method | Mesh |   PSNR    |   LPIPS   |   SSIM    |
> | :----: | :--: | :-------: | :-------: | :-------: |
> |  Ours  |  GT  | **26.14** | **0.869** | **0.161** |
> |  Ours  | ACR  |   25.83   |   0.864   |   0.167   |
>
> The performance of our methods with parameters estimated by ACR is still satisfactory. This is because we have added regulation to texture map bias to prevent our model from over-fitting the incorrect alignment between hand appearance and geometry, which addresses minor estimation errors effectively.
>
> (ii) To validate the impact of using masks produced by SAM, we conduct ablation experiments between using the SAM masks and the mesh rendering masks. The qualitative results  are shown in RF-Fig. 2 Ablaiton Mask and the quantitative results are listed as follows:
>
> | Method | Mask |   PSNR    |   SSIM    |   LPIPS   |
> | :----: | :--: | :-------: | :-------: | :-------: |
> |  Ours  | SAM  | **26.52** | **0.888** | **0.135** |
> |  Ours  | mesh |   25.99   |   0.877   |   0.142   |
>
> We can see that using the masks produced by SAM improves the model performance, due to the better alignment with hand boundaries,  which helps to mitigate background clutters.
>
> **W2 Non-smooth hand boundaries**: Thank you for pointing this out. Compared to NeRF-based methods, GS-based methods usually use fewer rendering points, which may lead to less smooth hand boundaries. To confirm this, we further lower the number of Gaussian points to 24k in our method. The experimental results are shown in RF-Fig. 2 (denoted as Low G. Num. in Ablation S1). We find that hand boundaries indeed deteriorate due to the fewer number of Gaussian points.
>
> **Q1 The advantage of GS compared to high-resolution hand meshes**: The advantage of using GS is that GS is more flexible compared to parametric high-resolution hand meshes. The geometry of parametric high-resolution hand meshes is relatively fixed and
> thus it's hard for them to present fine-grained geometric deformation of interacting hands. To alleviate such limitation of high-resolution hand mesh, we devise a Gaussian refinement module that adaptively adjusts the number and positions of 3D Gaussian according to different hand poses and identities, which results in better hand geometry in the rendered images.
>
> **Q2 Comparing the Reconstruction Time with OHTA**: The one-shot avatar reconstruction of OHTA takes 56 minutes on an A100 GPU (according to the OHTA paper) while our method takes 2.5 minutes on an A6000 GPU. For the fairness of comparison, we conduct an experiment to test the fine-tuning cost of our method and OHTA, of which the results are listed below:
>
> | Method |   PSNR    |   SSIM    |   LPIPS   | Fine Tuning Time |
> | :----: | :-------: | :-------: | :-------: | :--------------: |
> |  Ours  | **26.14** | **0.869** |   0.161   |   2.5 minutes    |
> | OHTA*  |   25.31   |   0.851   |   0.184   |    5 minutes     |
> | OHTA** |   25.96   |   0.864   | **0.160** |    25 minutes    |
>
> where OHTA* means that it uses the same number of training steps as in our method, while that of OHTA** is times by five to ensure sufficient fitting. From these results, we can see that the computational complexity of our method is significantly lower than that of OHTA. The reduction of fine-tuning consumption comes from three aspects: (i) We choose to fine-tune the texture map bias instead of the whole network, which accelerates the process and lowers the calculation cost. (ii) OHTA designs a two-stage fine-tuning method while we only have one fine-tuning stage as texture map bias can be easily regularized. (iii) Thanks to the design of Gaussian Splatting, we lower the rendering points and reduce the rendering cost compared to NeRF-based methods.

---

> > ### Author Response · Authors · 2024-08-12
> >
> > Dear Reviewer mjNr,
> >
> > We have tried to address your concerns in our earlier responses. We are looking forward to your suggestions. May we know if our rebuttals answer all your questions? We truly appreciate it.

---

> > > ### Comment · Reviewer_mjNr · 2024-08-13
> > >
> > > I appreciate the authors' efforts to address my concerns. Most of my questions have been resolved, so I raised my score.

---

> > > > ### Author Response · Authors · 2024-08-14
> > > >
> > > > Dear Reviewer mjNr,
> > > >
> > > > Thank you for your appreciation. Your suggestions and feedback truly contribute to the improvement of our paper.  We will ensure that all revisions are included in the final version.

---

### Official Review · Reviewer_eWJx · 2024-07-11

**Soundness:** 3
**Presentation:** 2
**Contribution:** 2
**Rating:** 5
**Confidence:** 4

**Summary:**

The paper proposes an approach to achieve one-shot interacting hand avatar reconstruction via 3D Gaussian Splatting. The authors design a two-stage framework: the first stage learns learning-based features and optimization-based identity maps, and the second stage performs one-shot reconstruction by optimizing only the identity map. To better capture the relationship between the two hands, they devise an interaction-aware attention module. Moreover, they design a self-adaptive GS density control approach for this task. The experiments demonstrate the effectiveness of their designs and show that the approach achieves state-of-the-art performance.

**Strengths:**

1. The paper presents the first framework capable of achieving one-shot animatable two-hand avatars.
2. The disentangled designs for one-shot reconstruction, using optimization-based identity maps for the one-shot stage, are technically sound.

**Weaknesses:**

1. For interacting hands, significant shadows occur between the two hands. The proposed method does not account for shadow modeling, which overlooks the dynamic nature of the hands and makes the approach less practical for real-world scenarios to some extent.
2. The experimental results are not comprehensive.
- It would be beneficial to present some in-the-wild testing results to demonstrate the method's plausibility for real applications. This would also justify the design for addressing out-of-distribution (OOD) data.
- Including some failure cases would help to illustrate the boundaries of the work.

**Questions:**

1. Why should we use interacting hands for avatar creation? What are the benefits compared to single-hand modeling?
2. What is the detailed implementation of OHTA? The results appear to be worse than the reported results.
3. How does the accuracy of mesh reconstruction impact the final results?

**Limitations:**

The authors present the "Limitations and Society Impact" section.

---

> ### Author Rebuttal · Authors · 2024-08-06
>
> We sincerely thank you for your suggestive comments. We have performed multiple experiments and presented additional real-world results to further validate the proposed method. Our point-by-point responses are listed as follows:
>
> **W1 Shadow modeling**: Thank you for your suggestive advice. Our current framework indeed focuses on the disentanglement of poses and identities and ignores shadow modeling. However, as we do not impose any constraint on shadows, our framework is of great flexibility and hence is easy to incorporate existing shadow models. To demonstrate this, we follow OHTA to disentangle pose-related features and predict shadow coefficients, which allows our method to model shadows and albedos separately. As shown in RF-Fig. 2, our method with this enhancement successfully separates shadows caused by occlusions. Moreover, we also provide more in-the-wild examples regarding three challenging tasks in RF-Fig. 1. These examples indicate that even our current model can handle real-world scenarios. From these two aspects, we believe the proposed method is a feasible solution for various applications.
>
> **W2 In-the-wild results**: Thank you for your advice. As shown in RF-Fig. 1, we demonstrate more visual examples of three scenarios, including in-the-wild reconstruction, text-to-avatar, and texture editing. In the in-the-wild results, we use ACR to estimate hand pose and camera parameters from real-captured images. These results clearly show that our method can be applied to in-the-wild images and obtain considerable results. These results will be added to our revised paper.
>
> **W3 Failure cases**: We add the demonstration of failure cases as shown in RF-Fig. 3. If severe hand mesh estimation errors occur, our method may fail to model the proper appearance. We notice previous methods also suffer from this issue (as mentioned in the OHTA and VANeRF paper), and a better mesh recovery method is necessary in this case.
>
> **Q1 The benefits of using interacting hands for avatar creation**: The advantage of using two-hand images is that they provide more complementary information compared with single hands, especially when severe inter-hand occlusions occur. To validate this, we compare the performance of our method with that of the single-hand counterpart by masking one hand in reference images. The quantitative results are listed as follows:
>
> |   Method   |   PSNR    |   SSIM    |   LPIPS   |
> | :--------: | :-------: | :-------: | :-------: |
> |    Ours    | **26.14** | **0.869** | **0.161** |
> | Ours-Right |   24.81   |   0.852   |   0.178   |
> | Ours-Left  |   25.58   |   0.858   |   0.172   |
>
> The above results show that our method with two-hand images outperforms that with single-hand images significantly. Moreover, RF-Fig. 2 Ablation Hand Num. demonstrates the visual examples of this experiment. These examples also suggest that using both hands increases the quality of rendered images.
>
> Compared to single-hand modeling, the intra- and inter-hand interaction exacerbates information loss and introduces complex geometric deformations. It is necessary to study the interacting hand reconstruction. Extensive efforts have been made by related research communities to reconstruct interacting hands from a single image, such as VANeRF, ACR, and IntagHand. To improve the performance of interacting hand reconstruction, we propose an interaction-aware attention module and a self-adaptive Gaussian refinement module. These modules enhance image rendering quality in areas with intra- and inter-hand interactions.
>
> Furthermore, our method can construct a hand avatar using images of both hands with interaction or single-hand images. Since there are situations in which we only have single-hand images or interacting hand images, our method has a wider application than single-hand reconstruction methods.
>
> **Q2 Implementation of OHTA***: Please note that OHTA is a single-hand reconstruction approach, and **its setup and dataset completely differ from ours**. Because its pre-trained model cannot be used in our scenario, we utilize our own pre-trained model and only compare the fine-tuning stage design. The code of OTHA is not published before the deadline for paper submission, hence we implement OHTA* with the texture inversion stage and texture fitting stage following its paper. In the texture-inversion stage, we keep the weights of the pre-trained network frozen and optimize the identity code along with per-channel color calibration coefficients to fit the input image. In the texture-fitting stage, we fine-tune the MLP for texture feature extraction and impose the constraint that the texture-fitting results of reference views should be close to the rendering results before texture-fitting.
>
> Concerning the inferior performance of OHTA*, we consider that the previous number of training steps might be insufficient for OHTA* to fully fit reference images. To investigate this, we further prolong the training steps (denoted as OHTA**) and obtain better performance as follows:
>
> | Method |   PSNR    |   SSIM    |   LPIPS   | Fine Tuning Time |
> | :----: | :-------: | :-------: | :-------: | :--------------: |
> |  Ours  | **26.14** | **0.869** |   0.161   |   2.5 minutes    |
> | OHTA*  |   25.31   |   0.851   |   0.184   |    5 minutes     |
> | OHTA** |   25.96   |   0.864   | **0.160** |    25 minutes    |
>
> Yet, our method still achieves a better balance between rendering quality and fine-tuning time.

---

> > ### Comment · Reviewer_eWJx · 2024-08-12
> >
> > Thank you for your response. I appreciate the authors' efforts in addressing my concerns, but they remain unresolved.
> > Regarding shadow modeling, while the authors demonstrate the proposed method's ability to model shadows in the InterHand2.6M dataset, they have not shown its effectiveness in capturing interacting shadows in in-the-wild testing data. I recommend that the authors present visually appealing examples of interacting shadows in these in-the-wild results.
> > Additionally, the in-the-wild visuals are not convincing. I suggest using higher-quality images and improved presentation methods to better showcase the results.

---

> > > ### Author Response · Authors · 2024-08-13
> > >
> > > Dear Reviewer eWJx,
> > >
> > > Thanks for suggesting improving the representation of shadow modeling for in-the-wild images. Although shadow modeling is beyond the scope of this paper (as we don't have ground-truth albedo images for explicit supervision), we provide additional examples of this in the following anonymous link:
> > >
> > > https://anonymous.4open.science/r/in_the_wild_shadow-35EB/README.md.
> > >
> > > Please have a look at these examples, where we show in-the-wild inputs (left) and the synthesized images (right-top), albedo images (right-middle), and shadow images (right-bottom) under different poses. We also label shadow areas caused by occlusions in red rectangles, which clearly show that our method generates appealing interacting shadows of interacting hands.
> > >
> > > We sincerely hope this can address your concern and help to improve your rating of our paper.

---

> > > > ### Comment · Reviewer_eWJx · 2024-08-13
> > > >
> > > > I appreciate the authors' efforts in responding to the feedback. While the response addresses some of my concerns, I continue to suggest that the authors present more visually appealing results in the revised version to better demonstrate their method's effectiveness. Furthermore, I strongly recommend that the authors incorporate the suggestions from all reviewers to enhance the overall completeness of the paper. Based on the current revision, I am inclined to rate this paper as Borderline Accept.

---

> > > > > ### Author Response · Authors · 2024-08-13
> > > > >
> > > > > Dear Reviewer eWJx,
> > > > >
> > > > > Thank you very much for your appreciation. Your suggestions and comments indeed help to improve our paper greatly. We will include more visual results (especially for the in-the-wild images) and all other revisions in our paper.

---

> ### Author Response · Authors · 2024-08-07
>
> **Q3 The impact of the accuracy of mesh reconstruction**: We conduct an experiment to estimate the effects of different meshes. In this experiment, we compare the performance of our method with the ground-truth meshes provided by the Interhand2.6M dataset, and that with meshes predicted by ACR. The qualitative results are shown in RF-Fig. 2 Ablaiton Mesh and the quantitative results are listed as follows:
>
> | Method | Mesh |   PSNR    |   LPIPS   |   SSIM    |
> | :----: | :--: | :-------: | :-------: | :-------: |
> |  Ours  |  GT  | **26.14** | **0.869** | **0.161** |
> |  Ours  | ACR  |   25.83   |   0.864   |   0.167   |
>
> The performance of our methods with parameters estimated by ACR is still satisfactory. This is because we have added regulation to texture map bias to prevent our model from over-fitting the incorrect alignment between hand appearance and geometry, which addresses minor estimation errors effectively.

---

> > ### Author Response · Authors · 2024-08-12
> >
> > Dear Reviewer eWJx,
> >
> > Thank you for taking the time to provide valuable suggestions. We have made our best efforts to address your concerns. If you have any other questions or suggestions, we are very grateful to discuss them with you.

---

### Official Review · Reviewer_gWjw · 2024-07-12

**Soundness:** 3
**Presentation:** 3
**Contribution:** 3
**Rating:** 6
**Confidence:** 4

**Summary:**

This paper proposes a novel two-stage interaction-aware GS framework to create animatable avatars for interacting hands from single-image inputs. The proposed method disentangle the 3D presentation of hands into optimization-based identity maps and learning-based latent geometric features and neural texture maps, and exploits cross-subject hand priors and refines 3D Gaussians in interacting areas. Experiments are conducted on the Interhand2.6M dataset with state-of-the-art method comparisons.

**Strengths:**

1) The paper is well written and easy to follow, and it is technically sound to disentangle geometric and texture feature learning to achieve prior learning across different subjects.
2) The idea of the Interaction-aware Attention module is novel, which detect the interation-aware geometric deformation in an intuitive way.
3) The quantitative and qualitative results are of good quality and surpass the current SOTA on Interhand2.6M, and the ablation studies are quite extensive.
4) Comprehensive qualitative ablation studies are shown. The authors additionally present qualitative results on the interaction areas, showing enhanced rendering quality.

**Weaknesses:**

1) The data flow in Figure 3 is somewhat unclear. For example, what does the dotted line from the Interaction Points to the Interaction-Aware Attention present? In addition, using different colored sub boxes for geometric flow and texture flow can help readers more intuitively understand the disentanglement of hand priors.
2) In line 119 of page 4, there is a typo error. 'Sec.' is missing.
3) The generalization of in-the-wild images has not been demonstrated.

**Questions:**

1) In lines 85-87 of page 3, why can disentangling hand prior learning reduce the cost of one-shot fitting？Is this related to introducing texture map bias to accelerate one shot fitting
2) Why is the interacting label $d(q)$ binarized, rather than applied by soft activation?
3) An important baseline, OHTA, was re-implemented and compared in the experiment. Can you provide more details?

**Limitations:**

Yes

---

> ### Author Rebuttal · Authors · 2024-08-06
>
> We are grateful for your acknowledgment of the technical contributions and novelty of our method. We have considered your as well as other reviewers' suggestions carefully and tried to solve them as follows:
>
> **W1&W2 Presentation**: Thank you for pointing this out. We will enhance our presentation in Figure 3 and revise the typo accordingly. The dotted line from the Interaction Points to the Interaction-Aware Attention means that we feed the information of interaction points into the Interaction-Aware Attention module. Considering the page limitation, the revised version of Figure 3 using different colored sub-boxes for geometric flow and texture flow will be presented in the future version.
>
> **W3 In-the-wild results**: RF-Fig. 1 shows more real-world results and applications of our proposed method, affirming its effectiveness in real-world scenarios and its suitability for handling out-of-distribution data. RF-Fig. 1 comprises three challenging tasks, i.e., text-to-avatar, real image in-the-wild reconstruction, and texture editing. The proposed method handles these tasks effectively.
>
> **Q2 The cost of one-shot fitting**: Yes, introducing texture map bias helps accelerate one-shot fitting due to the reduction of learnable parameters. Compared to directly fine-tuning the network (427 M), optimizing the texture map bias (0.65M) reduces the fine-tuning time significantly.
>
> **Q2 Interacting label**: We adopt binarized interacting labels instead of utilizing probabilistic ones because we propose a simple and effective strategy to detect interacting points (P6, L193-197). Specifically, we calculate the difference between the neighboring point sets of posed hand meshes and a canonical mesh. If the difference is above the user-defined threshold, the corresponding interacting label is set to 1 otherwise 0. In our experiments, we find this strategy effective enough to detect self-interacting and cross-interacting points.
>
> **Q3 Implementation of OHTA**: Because the code of OTHA is not published before the deadline for paper submission and the pre-trained model of OTHA cannot be used in our scenario, we utilize our own pre-trained model and only compare the fine-tuning stage design. Specifically, we implement OHTA* with the texture inversion stage and texture fitting stage following its paper. In the texture-inversion stage, we keep the weights of the pre-trained network frozen and optimize the identity code along with per-channel color calibration coefficients to fit the input image. In the texture-fitting stage, we fine-tune the MLP for texture feature extraction and impose the constraint that the texture-fitting results of reference views should be close to the rendering results before texture-fitting.
>
> Concerning the inferior performance of OHTA*, we consider that the previous number of training steps might be insufficient for OHTA* to fully fit the reference image. To investigate this, we further prolong the training steps (denoted as OHTA**) and obtain better performance as follows:
>
> | Method |   PSNR    |   SSIM    |   LPIPS   | Fine Tuning Time |
> | :----: | :-------: | :-------: | :-------: | :--------------: |
> |  Ours  | **26.14** | **0.869** |   0.161   |   2.5 minutes    |
> | OHTA*  |   25.31   |   0.851   |   0.184   |    5 minutes     |
> | OHTA** |   25.96   |   0.864   | **0.160** |    25 minutes    |
>
> Yet, our method still achieves a better balance between rendering quality and fine-tuning time.

---

> > ### Author Response · Authors · 2024-08-12
> >
> > Dear Reviewer gWjw,
> >
> > We are grateful for your previous comments and suggestions. We have responded to your comments in earlier days. If you have any other questions or comments, we will be very happy to discuss them with you.

---

> > ### Comment · Reviewer_gWjw · 2024-08-12
> >
> > The author provided the in-the-wild qualitative results, and more detailed comparisons of the baseline method OHTA, which to some extent addressed the previous concerns. Therefore, I decide to maintain the initial rating, Weak Accept.

---

> > > ### Author Response · Authors · 2024-08-13
> > >
> > > Dear Reviewer gWjw,
> > >
> > > Thank you very much for your time and appreciation. All the revisions (in-the-wild qualitative results and detailed comparisons) will be included in our revised paper.

---

> ### Comment · Reviewer_NUjU · 2024-08-13
>
> I appreciate the authors' proactive response. The supplementary ablation experiments on camera parameters and the additional experiments using predicted meshes instead of GT meshes have addressed my concerns.
>
> Regarding OHTA and the authors' reproduced versions OHTA* or OHTA**, I suggest that the authors clarify their descriptions in the revision. Based on my understanding, the actual OHTA is not equivalent to OHTA* (Inv+Calib), as there are other details such as the use of reference views for assistance during finetune.
>
> I'd also like to gently remind the authors about their commitment in the "Open access to data and code" section, where they mentioned that "Code and models will be released upon acceptance." Following through on this would be greatly appreciated and could help address any lingering concerns about reproducibility that reviewers might have.
>
> At this point, I am inclined to give this paper a score of Borderline Accept.

---

### Official Review · Reviewer_NUjU · 2024-07-13

**Soundness:** 2
**Presentation:** 3
**Contribution:** 2
**Rating:** 5
**Confidence:** 5

**Summary:**

- The author extends the concept of one-shot hand avatar creation from the single hand in the previous OHTA paper to two hands.
- The author proposes a novel two-stageGS framework for the reconstruction and rendering of avatars. This framework utilizes 2D identity maps to represent identity information and assist in the learning of neural textures. Additionally, an attention mechanism and a Gaussian refinement module are designed to handle the interaction information.
- The author demonstrates the superior performance of their method compared to the baseline on the Interhand2.6M dataset.

**Strengths:**

- The author has successfully applied the Gaussian splatting module for the first time in the problem of one-shot hand avatar creation.

- The method proposed in the paper has shown improved performance on the Interhand dataset.

- The paper is easy to follow.

**Weaknesses:**

- The article represents an incremental work.

- Initially, it is not convincing why one would construct a hand avatar using images of both hands with interaction instead of building them separately. What are the advantages?

- Secondly, the advantage of the one-shot method lies in addressing in-the-wild scenarios. The pioneering work in the single hand direction, OHTA, demonstrated a multitude of in-the-wild results, but this paper does not. This raises doubts about the effectiveness of this method in in-the-wild scenarios.

- The method's modeling of the hand is not sufficiently detailed. Previous methods such as Handavatar, OHTA, etc., have considered shadows caused by occlusions, but it is not clear how this article's method learns such priors.

**Questions:**

- The OHTA* method compared in the article seems more like an ablation of the full model. The performance of the OHTA algorithm reproduced by the authors is worse than in the original paper. Can an explanation be provided on how this baseline was implemented, and can some reasons for the poorer performance be analyzed?

- The paper incorporates camera intrinsic and extrinsic parameters as part of the Geometric Encoding. What are the advantages of introducing this information? Is there any related ablation analysis?

- Is the initial mesh of the bimanual results shown by the author estimated or taken from the dataset's ground truth? If it is estimated, what estimation method was used? How is the error in pose estimation addressed?

**Limitations:**

The author has already addressed the limitations of the method.

---

> ### Author Rebuttal · Authors · 2024-08-06
>
> We sincerely thank the reviewer’s valuable comments and suggestions. We have conducted extensive ablation studies and provide more in-the-wild results to further verify the proposed method. Our responses are listed as follows:
>
> **W1 Contribution and novelty**: We feel it is necessary to clarify the technical contributions of the proposed method from the following perspectives:
>
> First, constructing two-hand avatars via simple extensions of one-hand baselines faces certain limitations. Particularly, it is challenging for single-hand methods to handle severe occlusions and inter-hand interference, which results in inferior performance. This can be validated by our new quantitative (the table in our following response to W2) and qualitative comparisons (RF-Fig. 2, Ablation Hand Num.) provided in the rebuttal. Moreover, single-hand methods do not include modules to leverage the structure and texture similarity of hands, which are informative in various scenarios. Therefore, extensive methods (VANeRF[1], ACR[2], and Im2hands[3]) are tailored specially for the two-hand task.
>
> Second, three novel components have been introduced in our method, including (i) the disentangled 3D presentation to improve flexibility and exploit data priors, (ii) the interaction-aware attention module to handle complex interactions, and (iii) a Gaussian refinement module to improve rendered image quality. The effectiveness of these modules has been validated by the ablation studies in our paper.
>
> Last but not least, we are glad that all other reviewers found the proposed method novel and effective, such as ``novel and intuitive" (Reviewer #gWjw), technically sound (Reviewer #eWJx and #mjNr), and one of the first works" (Reviewer #eWJx and #mjNr). We believe such appreciation can indicate that the proposed method is not incremental.
>
> Hence, we sincerely hope the novel and contributions of our method can be reconsidered. We are open to further discussion and will try our best efforts to address your concern.
>
> [1]Xuan Huang et al., 3d visibility-aware generalizable neural radiance fields for interacting hands. In Proceedings of the AAAI Conference on Artificial Intelligence, volume 38, pages 2400–2408, 2024.
>
> [2]Zhengdi Yu et al., Acr: Attention collaboration-based regressor for arbitrary two-hand reconstruction. In Proceedings of the IEEE/CVF Conference on Computer Vision and Pattern Recognition, pages 12955–12964, 2023.
>
> [3]Jihyun Lee et al., Im2hands: Learning attentive implicit representation of interacting two-hand shapes. In Proceedings of the IEEE/CVF Conference on Computer Vision and Pattern Recognition, pages 21169–21178, 2023.
>
> **W2 The advantages of using images of both hands**: The advantage of using two-hand images is that they provide more complementary information compared with those of single hands, especially when severe inter-hand occlusions occur. To validate this, we compare the performance of our method with that of the single-hand counterpart by masking one hand in reference images. The quantitative results are listed as follows:
>
> |   Method   |   PSNR    |   SSIM    |   LPIPS   |
> | :--------: | :-------: | :-------: | :-------: |
> |    Ours    | **26.14** | **0.869** | **0.161** |
> | Ours-Right |   24.81   |   0.852   |   0.178   |
> | Ours-Left  |   25.58   |   0.858   |   0.172   |
>
> The above results show that our method with two-hand images outperforms that with single-hand images significantly. Moreover, RF-Fig. 2 Ablation Hand Num. demonstrates the visual examples of this experiment. These examples also suggest that using both hands increases the quality of rendered images.
>
> Compared to single-hand modeling, the intra- and inter-hand interaction exacerbates information loss and introduces complex geometric deformations. It is necessary to study the interacting hand reconstruction. Extensive efforts have been made by related research communities to reconstruct interacting hands from a single image, such as VANeRF, ACR, and IntagHand. To improve the performance of interacting hand reconstruction, we propose an interaction-aware attention module and a self-adaptive Gaussian refinement module. These modules enhance image rendering quality in areas with intra- and inter-hand interactions.
>
> Furthermore, our method can construct a hand avatar using images of both hands with interaction or single-hand images. Since there are situations in which we only have single-hand images or interacting hand images, our method has a wider application than single-hand reconstruction methods.
>
> **W3 In-the-wild results**: As shown in RF-Fig. 1, We demonstrate more visual results of various applications, including in-the-wild results from real-captured images, text-to-avatar, and texture editing. For text-to-avatar, we utilize ControlNet with depth maps as condition information for hand image generation. For in-the-wild results, we use ACR to estimate hand pose and camera parameters from real-captured images. These results clearly show that our method can be applied to in-the-wild images and obtain considerable results.
>
> **W4 Shadow modeling**: Handavatar and OHTA disentangle the pose-related information and further predict the shadow coefficient so that they can visualize the shadow and albedo separately. We disentangle pose-related information from identity-related information to ensure generalization ability. Instead of predicting the shadow coefficient, we render the shadow and albedo simultaneously. However, our method is flexible in choosing shadow modeling strategies. To demonstrate this, we conduct an experiment that uses disentangled pose-related features to predict shadow coefficients as in OHTA, allowing for the separate modeling of shadows and albedos. The visual results are shown in RF-Fig. 2 Shadow. From these results, it is apparent that our method successfully separates the shadows (e.g. the area beneath the palm). The qualitative results are shown in RF-Fig. 2 w/ Shadow of Ablation S1.

---

> ### Author Response · Authors · 2024-08-06
> **Our Responses to Questions**
>
> **Q1 Implementation of OHTA***: Please note that OHTA is a single-hand reconstruction approach, and **its setup and dataset completely differ from ours**. Because its pre-trained model cannot be used in our scenario, we utilize our own pre-trained model and only compare the fine-tuning stage design. The code of OTHA is not published before the deadline for paper submission, hence we implement OHTA* with the texture inversion stage and texture fitting stage following its paper. In the texture-inversion stage, we keep the weights of the pre-trained network frozen and optimize the identity code along with per-channel color calibration coefficients to fit the input image. In the texture-fitting stage, we fine-tune the MLP for texture feature extraction and impose the constraint that the texture-fitting results of reference views are close to the rendering results before texture-fitting.
>
> Since our fine-tuning process only contains one stage while OHTA contains two fine-tuning stages, we use twice the training steps for OHTA* (one for each stage). Concerning the inferior performance of OHTA*, we consider that the previous number of training steps might be insufficient for OHTA* to fully fit the reference image. To investigate this, we further prolong the training steps (denoted as OHTA**) and obtain better performance as follows:
>
> | Method |   PSNR    |   SSIM    |   LPIPS   | Fine Tuning Time |
> | :----: | :-------: | :-------: | :-------: | :--------------: |
> |  Ours  | **26.14** | **0.869** |   0.161   |   2.5 minutes    |
> | OHTA*  |   25.31   |   0.851   |   0.184   |    5 minutes     |
> | OHTA** |   25.96   |   0.864   | **0.160** |    25 minutes    |
>
> Yet, our method still achieves a better balance between rendering quality and fine-tuning time.
>
> **Q2 The advantages of introducing camera parameters**: Thank you for your advice. We conduct the related ablation experiments to analyze the impact of the camera parameters. The qualitative results are shown in RF-Fig. 2 (denoted as w/o Cam. in Ablaiton S1) and the quantitative results (denoted as w/o Cam.) are listed as follows:
>
> |  Method  |   PSNR    |   SSIM    |   LPIPS   |
> | :------: | :-------: | :-------: | :-------: |
> |   Ours   | **28.11** | **0.902** | **0.130** |
> | w/o Cam. |   25.91   |   0.862   |   0.198   |
>
> The performance of the baseline without camera parameters drops obviously. Considering the anisotropy properties of Gaussian points (e.g. scale and opacity), it is necessary to utilize camera parameters to encode the view direction information for rendering.
>
> **Q3 Mesh estimation method**: We use the ground-truth hand parameters provided by the dataset. We conduct the experiment using hand pose parameters estimated by ACR. The qualitative results are shown in RF-Fig. 2 Ablation Mesh and the quantitative results are listed as follows:
>
> | Method | Mesh |   PSNR    |   LPIPS   |   SSIM    |
> | :----: | :--: | :-------: | :-------: | :-------: |
> |  Ours  |  GT  | **26.14** | **0.869** | **0.161** |
> |  Ours  | ACR  |   25.83   |   0.864   |   0.167   |
>
> The performance of our methods with parameters estimated by ACR is still satisfactory. This is because we have added regulation to the texture map bias in order to prevent the model from over-fitting to the incorrect alignment relation between hand appearance and geometry, which addresses minor estimation errors effectively. However, as demonstrated in the failure cases in RF-Fig. 3, serious estimation errors make it difficult for our model to learn the correct appearance. Previous works like OHTA also mention this issue, which might be alleviated by more reliable hand regressors.

---

> > ### Author Response · Authors · 2024-08-12
> >
> > Dear Reviewer NUjU,
> >
> > We have tried our best to address all the concerns in our earlier responses. We are very happy to discuss any additional questions or suggestions you have.
> >
> > Thank you for your time and consideration.

---

> > > ### Author Response · Authors · 2024-08-13
> > >
> > > Dear Reviewer NUjU,
> > >
> > > Thank you for increasing the score of our paper.  All the revisions will be incorporated into our paper. The link to the code and models will be provided in our revised paper to facilitate the following research.

---

### Author Rebuttal · Authors · 2024-08-06

First, we thank ACs for organizing such a wonderful reviewing process and reviewers for their constructive comments that help to improve our paper greatly. We appreciate the confirmations from the reviewers on the proposed method, including Reviewer #NUjU "successfully applied the Gaussian splatting module for **the first time** in the problem of one-shot hand avatar creation", Reviewer #gWjw "The idea of the Interaction-aware Attention module is **novel**, which detect the interaction-aware geometric deformation in an intuitive way.", Reviewer #eWJx "The disentangled designs for one-shot reconstruction, using optimization-based identity maps for the one-shot stage, are **technically sound**.", and Reviewer #mjNr "**Sufficient technical novelty**". Reviewers also found this paper well-written and easy to follow (Reviewer #NUjU, #gWjw, and #mjNr). We hope this paper can shed light on GS-based one-shot 3D interacting hand reconstruction and be a considerable baseline.

We understand that the major concerns of reviewers are in-the-wild results and more comprehensive ablation studies. Hence, we provide two extra visual comparisons in the rebuttal file (denoted as RF), which include examples of in-the-wild results of three different tasks and four ablation studies.

The two visual comparisons are summarized as follows:

**Fig. 1** shows multiple examples of the proposed method which validate the effectiveness of our method in in-the-wild scenarios and justify the design for addressing out-of-distribution data. Three in-the-wild tasks are considered in this figure, including text-to-avatar (first four rows), in-the-wild reconstruction from real images (5th to 8th row), and texture editing (last row).

**Fig. 2** (top) demonstrates visual examples of shadow disentanglement (top) and qualitative results of four ablation studies (bottom). The visualization of shadow disentanglement includes a shaded image, an albedo image, and a shadow image for each pair of hands. This figure shows the shadow (dark) areas clearly, which suggests our method can disentangle albedo and shadow.

**Fig. 2** (bottom) shows the results of four ablation studies, which can be further divided into the following four groups:

**1. Ablation study on single-hand images** (denoted as Ablation Hand Num.): We construct a single-hand baseline of the proposed method by masking one of the two hands in reference images. The results show that our method achieves better performance compared with the single-hand baseline, which is reasonable as two-hand images contain more complementary information.

**2. Ablation study on segmentation method** (Ablation Mask): To validate the impact of segmentation masks, we consider masks predicted by SAM and those rendered by ground-truth meshes. We observe that SAM masks enhance the performance of our model, since they are better aligned with hands and prevent background clutters.

**3. Ablation S1**: Ablation S1 shows the visual results of three different ablation settings: including (i) **the use of camera parameters**, which is a variant of our method without the camera parameters (denoted as w/o Cam.) and suffers from obvious performance drop; (ii) **adding shadow coefficient**, which incorporates shadow coefficient prediction into the proposed method (w/ Shadow); (iii) **Low Gaussian Points**, which reduces the number of Gaussian points to 24k to show that coarsen hand boundaries are caused by fewer numbers of Gaussian points.

**4. Ablation study on mesh estimation method** (Ablation Noise): We use hand parameters estimated by ACR to analyze how the accuracy of mesh reconstruction impacts the final results. The performance of our method with the predicted parameters is still satisfying, which suggests our method is robust to noise hand meshes to a certain extent.

For the convenience of reading, **the quantitative results of the above studies are provided separately in our comments to each reviewer**. With these results, we hope all concerns have been addressed successfully.

The above revision will be incorporated into our paper and we are looking forward to comprehensive discussions in the next few days.

---

### Decision · Program_Chairs · 2024-09-25

**Decision:**

Accept (poster)

**Comment:**

This is a borderline paper. Based on the initial ratings, and especially after the rebuttal, all reviewers are satisfied with the methodology, evaluation, and the responses provided by the authors.